# Tight conditions for when the NTK approximation is valid

**Enric Boix-Adsera** *eboix@mit.edu*
*MIT Electrical Engineering and Computer Science*
*Apple*

**Etai Littwin** *elittwin@apple.com*
*Apple*

**Reviewed on OpenReview:** *https://openreview.net/forum?id=qM7JPBYROr*

## Abstract

We study when the neural tangent kernel (NTK) approximation is valid for training a model with the square loss. In the lazy training setting of Chizat et al. (2019), we show that rescaling the model by a factor of $\alpha = O(T)$ suffices for the NTK approximation to be valid until training time $T$. Our bound is tight and improves on the previous bound of Chizat et al. (2019), which required a larger rescaling factor of $\alpha = O(T^2)$.

## 1 Introduction

In the modern machine learning paradigm, practitioners train the weights $\boldsymbol{w}$ of a large neural network model $f_{\boldsymbol{w}} : \mathbb{R}^{d_{in}} \to \mathbb{R}^{d_{out}}$ via a gradient-based optimizer. Theoretical understanding lags behind, since the training dynamics are non-linear and hence difficult to analyze. To address this, Jacot et al. (2018) proposed an approximation to the dynamics called the *NTK approximation*, and proved it was valid for infinitely-wide networks trained by gradient descent[1]. The NTK approximation has been extremely influential, leading to theoretical explanations for a range of questions, including why deep learning can memorize training data (Du et al., 2018; 2019; Allen-Zhu et al., 2019a;b; Arora et al., 2019a; Cao & Gu, 2019; Lee et al., 2019), why neural networks exhibit spectral bias (Cao et al., 2019; Basri et al., 2020; Canatar et al., 2021), and why different architectures generalize differently (Bietti & Mairal, 2019; Mei et al., 2021; Wang et al., 2021). Nevertheless, in practice the training dynamics of neural networks often diverge from the predictions of the NTK approximation (see, e.g., Arora et al. (2019b)) Therefore, it is of interest to understand exactly under which conditions the NTK approximation holds. In this paper, we ask the following question:

*Can we give tight conditions for when the NTK approximation is valid?*

### 1.1 The "lazy training" setting of Chizat et al. (2019)

The work of Chizat et al. (2019) showed that the NTK approximation actually holds for training any differentiable model, as long as the model's outputs are *rescaled* so that the model's outputs change by a large amount even when the weights change by a small amount. The correctness of the NTK approximation for infinite-width models is a consequence of this observation, because by the default the model is rescaled as the width tends to infinity; see the related work in Section 1.3 for more details.

**Rescaling the model** Let $h : \mathbb{R}^p \to \mathcal{F}$ be a smoothly-parameterized model, where $\mathcal{F}$ is a separable Hilbert space. Let $\alpha > 0$ be a parameter which controls the rescaling of the model and which should be thought of as large. We train the rescaled model $\alpha h$ with gradient flow to minimize a smooth loss function $R : \mathcal{F} \to \mathbb{R}_+$.[2]

---

[1]Under a specific scaling of the initialization and learning rate as width tends to infinity.

[2]We use the Hilbert space notation as in Chizat et al. (2019). We can recover the setting of training a neural network $f_{\boldsymbol{w}} : \mathbb{R}^d \to \mathbb{R}$ on a finite training dataset $\{(\boldsymbol{x}_1, y_1), \ldots, (\boldsymbol{x}_n, y_n)\} \subseteq \mathbb{R}^d \times \mathbb{R}$ with empirical loss function $\mathcal{L}(\boldsymbol{w}) = \frac{1}{n} \sum_{i=1}^n \ell(f_{\boldsymbol{w}}(\boldsymbol{x}_i), y_i)$ as follows. Let $\mathcal{H} = \mathbb{R}^n$ be the Hilbert space, let $h(\boldsymbol{w}) = [f_{\boldsymbol{w}}(\boldsymbol{x}_1), \ldots, f_{\boldsymbol{w}}(\boldsymbol{x}_n)]$, and let $R(\boldsymbol{v}) = \frac{1}{n} \sum_{i=1}^n \ell(v_i, y_i)$.

Namely, the weights $\boldsymbol{w}(t) \in \mathbb{R}^p$ are initialized at $\boldsymbol{w}(0) = \boldsymbol{w}_0$ and evolve according to the gradient flow

$$\frac{d\boldsymbol{w}}{dt} = -\frac{1}{\alpha^2}\nabla_{\boldsymbol{w}}R(\alpha h(\boldsymbol{w}(t))). \tag{1}$$

**NTK approximation**   Define the linear approximation of the model around the initial weights $\boldsymbol{w}_0$ by

$$\bar{h}(\boldsymbol{w}) = h(\boldsymbol{w}_0) + Dh(\boldsymbol{w}_0)(\boldsymbol{w} - \boldsymbol{w}_0), \tag{2}$$

where $Dh$ is the first derivative of $h$ in $\boldsymbol{w}$. Let $\bar{\boldsymbol{w}}(t)$ be weights initialized at $\bar{\boldsymbol{w}}(0) = \boldsymbol{w}_0$ that evolve according to the gradient flow from training the rescaled linearized model $\alpha\bar{h}$:

$$\frac{d\bar{\boldsymbol{w}}}{dt} = -\frac{1}{\alpha^2}\nabla_{\bar{\boldsymbol{w}}}R(\alpha\bar{h}(\bar{\boldsymbol{w}}(t))). \tag{3}$$

The NTK approximation states that

$$\alpha h(\boldsymbol{w}(t)) \approx \alpha\bar{h}(\bar{\boldsymbol{w}}(t)).$$

In other words, it states that the linearization of the model $h$ is valid throughout training. This allows for much simpler analysis of the training dynamics since the model $\bar{h}$ is linear in its parameters, and so the evolution of $\bar{h}(\bar{\boldsymbol{w}})$ can be understood via a kernel gradient flow in function space.

**When is the NTK approximation valid?**   Chizat et al. (2019) proves that if the rescaling parameter $\alpha$ is large, then the NTK approximation is valid. The intuition is that the weights do not need to move far from their initialization in order to change the output of the model significantly, so the linearization (2) is valid for longer. Since the weights stay close to initialization, Chizat et al. (2019) refer to this regime of training as "lazy training." The following bound is proved.[3] Here

$$R_0 = R(\alpha h(\boldsymbol{w}_0))$$

is the loss at initialization, and

$$\kappa = \frac{T}{\alpha}\text{Lip}(Dh)\sqrt{R_0},$$

is a quantity that will also appear in our main results.

**Proposition 1.1** (Theorem 2.3 of Chizat et al. (2019))**.** *Let $R(y) = \frac{1}{2}\|y - y^*\|^2$ be the square loss, where $y^* \in \mathcal{F}$ are the target labels. Assume that $h$ is $\text{Lip}(h)$-Lipschitz and that $Dh$ is $\text{Lip}(Dh)$-Lipschitz in a ball of radius $\rho$ around $\boldsymbol{w}_0$. Then, for any time $0 \le T \le \alpha\rho/(\text{Lip}(h)\sqrt{R_0})$,*

$$\|\alpha h(\boldsymbol{w}(T)) - \alpha\bar{h}(\bar{\boldsymbol{w}}(T))\| \le T\text{Lip}(h)^2\kappa\sqrt{R_0}. \tag{4}$$

Notice that as we take the rescaling parameter $\alpha$ to infinity, then $\kappa$ goes to 0, so the right-hand-side of (4) is small and the NTK approximation is valid.

## 1.2   Our results

Our contribution is to refine the bound of Chizat et al. (2019) for large time scales. We prove:

**Theorem 1.2** (NTK approximation error bound)**.** *Let $R(y) = \frac{1}{2}\|y - y^*\|^2$ be the square loss. Assume that $Dh$ is $\text{Lip}(Dh)$-Lipschitz in a ball of radius $\rho$ around $\boldsymbol{w}_0$. Then, at any time $0 \le T \le \alpha^2\rho^2/R_0$,*

$$\|\alpha h(\boldsymbol{w}(T)) - \alpha\bar{h}(\bar{\boldsymbol{w}}(T))\| \le \min(6\kappa\sqrt{R_0}, \sqrt{8R_0}). \tag{5}$$

Furthermore, the converse is true. Our bound is tight up to a constant factor.

**Theorem 1.3** (Converse to Theorem 1.2)**.** *For any $\alpha, T, \text{Lip}(Dh)$, and $R_0$, there is a model $h : \mathbb{R} \to \mathbb{R}$, an initialization $w_0 \in \mathbb{R}$, and a target $y^* \in \mathbb{R}$ such that, for the risk $R(y) = \frac{1}{2}(y - y^*)^2$, the initial risk is $R(\alpha h(w_0)) = R_0$, the derivative map $Dh$ is $\text{Lip}(Dh)$-Lipschitz, and*

$$\|\alpha h(w(T)) - \alpha\bar{h}(\bar{w}(T))\| \ge \min(\frac{1}{5}\kappa\sqrt{R_0}, \frac{1}{5}\sqrt{R_0}).$$

[3]See Section 1.3 for discussion on the other results of Chizat et al. (2019).

**Comparison to Chizat et al. (2019)** In contrast to our theorem, the bound (4) depends on the Lipschitz constant of $h$, and incurs an extra factor of $T\text{Lip}(h)^2$. So if $\text{Lip}(Dh)$, $\text{Lip}(h)$, and $R_0$ are bounded by constants, our result shows that the NTK approximation (up to $O(\epsilon)$ error) is valid for times $T = O(\alpha\epsilon)$, while the previously known bound is valid for $T = O(\sqrt{\alpha\epsilon})$. Since the regime of interest is training for large times $T \gg 1$, our result shows that the NTK approximation holds for much longer time horizons than previously known.

### 1.3 Additional related literature

**Other results of Chizat et al. (2019)** In addition to the bound of Proposition 1.1 above, Chizat et al. (2019) controls the error in the NTK approximation in two other settings: (a) for general losses, but $\alpha$ must be taken exponential in $T$, and (b) for strongly convex losses and infinite training time $T$, but the problem must be "well-conditioned." We work in the setting of Proposition 1.1 instead, since it is more aligned with the situation in practice, where we have long training times and the problem is ill-conditioned. Indeed, the experiments of Chizat et al. (2019) report that for convolutional neural networks on CIFAR10 trained in the lazy regime, the problem is ill-conditioned, and training takes a long time to converge.

**Other works on the validity of the NTK approximation** The NTK approximation is valid for infinitely-wide neural networks under a certain choice of hyperparameter scaling called the "NTK parametrization" Jacot et al. (2018). However, there is another choice of hyperparameter scaling, called the "mean-field parametrization", under which the NTK approximation is not valid at infinite width (Chizat & Bach, 2018; Rotskoff & Vanden-Eijnden, 2018; Sirignano & Spiliopoulos, 2022; Mei et al., 2018; 2019; Yang & Hu, 2021). It was observed by Chizat et al. (2019) that one can interpolate between the "NTK parametrization" and the "mean-field parametrization" by varying the lazy training parameter $\alpha$. This inspired the works Woodworth et al. (2020); Geiger et al. (2020; 2021), which study the effect of interpolating between lazy and non-lazy training by varying $\alpha$. Most work points towards provable benefits of non-lazy training (Allen-Zhu & Li, 2019; Bai & Lee, 2019; Bai et al., 2020; Chen et al., 2020; Nichani et al., 2022; Ghorbani et al., 2020; Malach et al., 2021; Abbe et al., 2022; 2023; Mousavi-Hosseini et al., 2022; Damian et al., 2022; Bietti et al., 2022; Ba et al., 2022) although interestingly there are settings where lazy training provably outperforms non-lazy training (Petrini et al., 2022).

Finally, our results do not apply to ReLU activations because we require twice-differentiability of the model as in Chizat et al. (2019). It is an interesting future direction to prove such an extension. One promising approach could be to adapt a technique of Cao & Gu (2020), which analyzes ReLU network training in the NTK regime by showing in Lemma 5.2 that around initialization the model is "almost" linear and "almost" smooth, even though these assumptions are not strictly met because of the ReLU activations.

## 2 Application to neural networks

The bound in Theorem 1.2 applies to lazy training of any differentiable model. As a concrete example, we describe its application to neural networks (a similar application was presented in Chizat et al. (2019)). We parametrize the networks in the mean-field regime, so that the NTK approximation is not valid even as the width tends to infinity. Therefore, the NTK approximation is valid only when we train with lazy training.

Let $f_{\boldsymbol{w}} : \mathbb{R}^d \to \mathbb{R}$ be a 2-layer network of width $m$ in the mean-field parametrization Chizat & Bach (2018); Rotskoff & Vanden-Eijnden (2018); Sirignano & Spiliopoulos (2022); Mei et al. (2018; 2019), with activation function $\sigma : \mathbb{R} \to \mathbb{R}$,

$$f_{\boldsymbol{w}}(\boldsymbol{x}) = \frac{1}{\sqrt{m}} \sum_{i=1}^{m} a_i \sigma(\sqrt{m}\langle \boldsymbol{x}_i, \boldsymbol{u}_i \rangle).$$

The weights are $\boldsymbol{w} = (\boldsymbol{a}, \boldsymbol{U})$ for $\boldsymbol{a} = [a_1, \ldots, a_m]$ and $\boldsymbol{U} = [\boldsymbol{u}_1, \ldots, \boldsymbol{u}_m]$. These are initialized at $\boldsymbol{w}_0$ with i.i.d. $\text{Unif}[-1/\sqrt{m}, 1/\sqrt{m}]$ entries. Given training data $(\boldsymbol{x}_1, y_1), \ldots, (\boldsymbol{x}_n, y_n)$, we train the weights of the network

with the mean-squared loss

$$\mathcal{L}(\boldsymbol{w}) = \frac{1}{n}\sum_{i=1}^{n} \ell(f_{\boldsymbol{w}}(\boldsymbol{x}_i), y_i), \quad \ell(a,b) = \frac{1}{2}(a-b)^2\,. \tag{6}$$

In the Hilbert space notation, we let $\mathcal{H} = \mathbb{R}^n$, so that the gradient flow training dynamics with loss (6) correspond to the gradient flow dynamics (1) with the following model and loss function

$$h(\boldsymbol{w}) = \frac{1}{\sqrt{n}}[f_{\boldsymbol{w}}(\boldsymbol{x}_1), \ldots, f_{\boldsymbol{w}}(\boldsymbol{x}_n)] \in \mathbb{R}^n, \quad R(\boldsymbol{v}) = \frac{1}{2}\|\boldsymbol{v} - \frac{\boldsymbol{y}}{\sqrt{n}}\|^2\,.$$

Under some regularity assumptions on the activation function (which are satisfied, for example, by the sigmoid function) and some bound on the weights, it holds that $\mathrm{Lip}(Dh)$ is bounded.

**Lemma 2.1** (Bound on $\mathrm{Lip}(Dh)$ for mean-field 2-layer network). *Suppose that there is a constant $K$ such that (i) the activation function $\sigma$ is bounded and has bounded derivatives $\|\sigma\|_\infty, \|\sigma'\|_\infty, \|\sigma''\|_\infty, \|\sigma'''\|_\infty \le K$, (ii) the weights have bounded norm $\|\boldsymbol{a}\| + \|\boldsymbol{U}\| \le K$, and (iii) the data points have bounded norm $\max_i \|\boldsymbol{x}_i\| \le K$. Then there is a constant $K'$ depending only $K$ such that*

$$\mathrm{Lip}(Dh) \le K'\,.$$

*Proof.* See Appendix C. □

Note that our bounds hold at any finite width of the neural network, because we have taken initialization uniformly bounded in the interval $[-1/\sqrt{m}, 1/\sqrt{m}]$. Since the assumptions of Theorem 1.2 are met, we obtain the following corollary for the lazy training dynamics of the 2-layer mean-field network.

**Corollary 2.2** (Lazy training of 2-layer mean-field network). *Suppose that the conditions of Lemma 2.1, and also that the labels are bounded in norm $\|\boldsymbol{y}\| \le \sqrt{nK}$. Then there are constants $c, C > 0$ depending only on $K$ such that for any time $0 \le T \le c\alpha^2$,*

$$\|\alpha h(\boldsymbol{w}(T)) - \alpha\bar{h}(\bar{\boldsymbol{w}}(T))\| \le C\min(T/\alpha, 1)\,.$$

Notice that training in the NTK parametrization corresponds to training the model $\sqrt{m}f_{\boldsymbol{w}}$, where $f_{\boldsymbol{w}}$ is the network in the mean-field parametrization. This amounts to taking the lazy training parameter $\alpha = \sqrt{m}$ in the mean-field setting. Therefore, under the NTK parametrization with width $m$, the bound in Corollary 2.2 shows that the NTK approximation is valid until training time $O(m)$ and the error bound is $O(T/\sqrt{m})$.

## 3 Proof ideas

### 3.1 Proof ideas for Theorem 1.2

**Proof of Chizat et al. (2019)** In order to give intuition for our proof, we first explain the idea behind the proof in Chizat et al. (2019). Define residuals $r(t), \bar{r}(t) \in \mathcal{F}$ under training the original rescaled model and the linearized rescaled model as $r(t) = y^* - \alpha h(\boldsymbol{w}(t))$ and $\bar{r}(t) = y^* - \alpha\bar{h}(\bar{\boldsymbol{w}}(t))$. It is well known that these evolve according to

$$\frac{dr}{dt} = -K_t r \quad \text{and} \quad \frac{d\bar{r}}{dt} = -K_0\bar{r}\,,$$

for the time-dependent kernel $K_t : \mathcal{F} \to \mathcal{F}$ which is the linear operator given by $K_t := Dh(\boldsymbol{w}(t))Dh(\boldsymbol{w}(t))^\top$. To compare these trajectories, Chizat et al. (2019) observes that, since $K_0$ is p.s.d.,

$$\frac{1}{2}\frac{d}{dt}\|r - \bar{r}\|^2 = -\langle r - \bar{r}, K_t r - K_0\bar{r}\rangle \le -\langle r - \bar{r}, (K_t - K_0)r\rangle\,,$$

which, dividing both sides by $\|r - \bar{r}\|$ and using that $\|r\| \leq \sqrt{R_0}$ implies

$$\frac{d}{dt}\|r - \bar{r}\| \leq \|K_t - K_0\|\|r\| \leq 2\text{Lip}(h)\text{Lip}(Dh)\|\boldsymbol{w} - \boldsymbol{w}_0\|\sqrt{R_0}\,. \tag{7}$$

Using the Lipschitzness of the model, Chizat et al. (2019) furthermore proves that the weight change is bounded by $\|\boldsymbol{w}(t) - \boldsymbol{w}_0\| \leq t\sqrt{R_0}\text{Lip}(h)/\alpha$. Plugging this into (7) yields the bound in Proposition 1.1,

$$\|\alpha h(\boldsymbol{w}(T)) - \alpha\bar{h}(\bar{\boldsymbol{w}}(T))\| = \|r(T) - \bar{r}(T)\| \leq 2\text{Lip}(h)^2\text{Lip}(Dh)R_0\alpha^{-1}\int_0^T t dt$$
$$= T^2\text{Lip}(h)^2\text{Lip}(Dh)R_0/\alpha\,.$$

**First attempt: strengthening of the bound for long time horizons**  We show how to strengthen this bound to hold for longer time horizons by using an improved bound on the movement of the weights. Consider the following bound on the weight change.

**Proposition 3.1** (Bound on weight change, implicit in proof of Theorem 2.2 in Chizat et al. (2019)).

$$\|\boldsymbol{w}(T) - \boldsymbol{w}_0\| \leq \sqrt{TR_0}/\alpha \quad and \quad \|\bar{\boldsymbol{w}}(T) - \boldsymbol{w}_0\| \leq \sqrt{TR_0}/\alpha\,. \tag{8}$$

*Proof of Proposition 3.1.* By (a) Cauchy-Schwarz, and (b) the nonnegativity of the loss $R$,

$$\|\boldsymbol{w}(T) - \boldsymbol{w}(0)\| \leq \int_0^T \|\frac{d\boldsymbol{w}}{dt}\| dt \overset{(a)}{\leq} \sqrt{T\int_0^T \|\frac{d\boldsymbol{w}}{dt}\|^2 dt} = \sqrt{-\frac{T}{\alpha^2}\int_0^T \frac{d}{dt}R(\alpha h(\boldsymbol{w}(t)))dt}$$
$$\overset{(b)}{\leq} \sqrt{TR_0}/\alpha\,.$$

The bound for $\bar{\boldsymbol{w}}$ is analogous. $\qquad\qquad\qquad\qquad\qquad\qquad\qquad\qquad\qquad\qquad\qquad\square$

This bound (8) has the benefit of $\sqrt{t}$ dependence (instead of linear $t$ dependence), and also does not depend on $\text{Lip}(h)$. So if we plug it into (7), we obtain

$$\|\alpha h(\boldsymbol{w}(T)) - \alpha\bar{h}(\bar{\boldsymbol{w}}(T))\| \leq 2\text{Lip}(h)\text{Lip}(Dh)R_0\alpha^{-1}\int_0^T \sqrt{t}dt$$
$$= \frac{4}{3}T^{3/2}\text{Lip}(h)\text{Lip}(Dh)R_0/\alpha\,.$$

This improves over Proposition 1.1 for long time horizons since the time dependence scales as $T^{3/2}$ instead of as $T^2$. However, it still depends on the Lipschitz constant $\text{Lip}(h)$ of $h$, and it also falls short of the linear in $T$ dependence of Theorem 1.2.

**Second attempt: new approach to prove Theorem 1.2**  In order to avoid dependence on $\text{Lip}(h)$ and obtain a linear dependence in $T$, we develop a new approach. We cannot use (7), which was the core of the proof in Chizat et al. (2019), since it depends on $\text{Lip}(h)$. Furthermore, in order to achieve linear $T$ dependence using (7), we would need that $\|\boldsymbol{w} - \boldsymbol{w}_0\| = O(1)$ for a constant that does not depend on the time horizon, which is not true unless the problem is well-conditioned.

In the full proof in Appendix A, we bound $\|r(T) - \bar{r}(T)\| = \|\alpha h(\boldsymbol{w}(T)) - \alpha\bar{h}(\bar{\boldsymbol{w}}(T))\|$, which requires working with a product integral formulation of the dynamics of $r$ to handle the time-varying kernels $K_t$ (Dollard & Friedman, 1984). The main technical innovation in the proof is Theorem A.8, which is a new, general bound on the difference between product integrals.

To avoid the technical complications of the appendix, we provide some intuitions here by providing a proof of a simplified theorem which does not imply the main result. We show:

**Theorem 3.2** (Simplified variant of Theorem 1.2). *Consider $r'(t) \in \mathcal{F}$ which is initialized as $r'(0) = r(0)$ and evolves as $\frac{dr'}{dt} = -K_T r'$. Then,*

$$\|r'(T) - \bar{r}(T)\| \leq \min(3\kappa\sqrt{R_0}, \sqrt{8R_0})\,. \tag{9}$$

It is at the final time $t = T$ that the kernel $K_t$ can differ the most from $K_0$. So, intuitively, if we can prove in Theorem 3.2 that $r'(T)$ and $\bar{r}(T)$ are close, then the same should be true for $r(T)$ and $\bar{r}(T)$ as in Theorem 1.2. For convenience, define the operators

$$A = Dh(\boldsymbol{w}_0)^\top \quad \text{and} \quad B = Dh(\boldsymbol{w}(T))^\top - Dh(\boldsymbol{w}_0)^\top .$$

Since the kernels do not vary in time, the closed-form solution is

$$r'(t) = e^{-(A+B)^\top(A+B)t}r(0) \quad \text{and} \quad \bar{r}(t) = e^{-A^\top At}r(0)$$

We prove that the time evolution operators for $r'$ and $\bar{r}$ are close in operator norm.

**Lemma 3.3.** *For any $t \geq 0$, we have $\|e^{-(A+B)^\top(A+B)t} - e^{-A^\top At}\| \leq 2\|B\|\sqrt{t}$.*

*Proof of Lemma 3.3.* Define $Z(\zeta) = -(A + \zeta B)^\top(A + \zeta B)t$. By the fundamental theorem of calculus

$$\|e^{-(A+B)^\top(A+B)t} - e^{-A^\top At}\| = \|e^{Z(1)} - e^{Z(0)}\| = \|\int_0^1 \frac{d}{d\zeta}e^{Z(\zeta)}d\zeta\| \leq \sup_{\zeta \in [0,1]} \|\frac{d}{d\zeta}e^{Z(\zeta)}\|.$$

Using the integral representation of the exponential map (see, e.g., Theorem 1.5.3 of (Dollard & Friedman, 1984)),

$$\|\frac{de^{Z(\zeta)}}{d\zeta}\| = \|\int_0^1 e^{(1-\tau)Z(\zeta)}(\frac{d}{d\zeta}Z(\zeta))e^{\tau Z(\zeta)}d\tau\|$$

$$= \|t\int_0^1 e^{(1-\tau)Z(\zeta)}(A^\top B + B^\top A + 2\zeta B^\top B)e^{\tau Z(\zeta)}d\tau\|$$

$$\leq \underbrace{\|t\int_0^1 e^{(1-\tau)Z(\zeta)}(A + \zeta B)^\top B e^{\tau Z(\zeta)}d\tau\|}_{\text{(Term 1)}} + \underbrace{\|t\int_0^1 e^{(1-\tau)Z(\zeta)}B^\top(A + \zeta B)e^{\tau Z(\zeta)}d\tau\|}_{\text{(Term 2)}} .$$

By symmetry under transposing and reversing time, (Term 1) = (Term 2), so it suffices to bound the first term. Since $\|e^{\tau Z(\zeta)}\| \leq 1$,

$$(\text{Term 1}) \leq t\int_0^1 \|e^{(1-\tau)Z(\zeta)}(A + \zeta B)^\top\|\|B\|\|e^{\tau Z(\zeta)}\|d\tau$$

$$\leq t\|B\|\int_0^1 \|e^{(1-\tau)Z(\zeta)}(A + \zeta B)^\top\|d\tau$$

$$= t\|B\|\int_0^1 \sqrt{\|e^{(1-\tau)Z(\zeta)}(A + \zeta B)^\top(A + \zeta B)e^{(1-\tau)Z(\zeta)}\|}d\tau$$

$$= \sqrt{t}\|B\|\int_0^1 \sqrt{\|e^{(1-\tau)Z(\zeta)}Z(\zeta)e^{(1-\tau)Z(\zeta)}\|}d\tau$$

$$\leq \sqrt{t}\|B\|\int_0^1 \sup_{\lambda \geq 0} \sqrt{\lambda e^{-2(1-\tau)\lambda}}d\tau$$

$$= \sqrt{t}\|B\|\int_0^1 \sqrt{1/(2e(1-\tau))}d\tau$$

$$= \sqrt{2t/e}\|B\| .$$

where in the third-to-last line we use the Courant-Fischer-Weyl theorem and the fact that $Z(\zeta)$ is negative semidefinite. Combining these bounds $\|e^{-(A+B)^\top(A+B)t} - e^{-A^\top At}\| \leq 2\sqrt{2t/e}\|B\| \leq 2\|B\|\sqrt{t}$. $\qquad\square$

Finally, let us combine Lemma 3.3 with the weight-change bound in Proposition 3.1 to prove Theorem 3.2. Notice that the weight-change bound in Proposition 3.1 implies

$$\|B\| \leq \text{Lip}(Dh)\|\boldsymbol{w}(T) - \boldsymbol{w}_0\| \leq \text{Lip}(Dh)\sqrt{TR_0}/\alpha\,.$$

So Lemma 3.3 implies

$$\|r'(T) - \bar{r}(T)\| \leq 2\text{Lip}(Dh)T\sqrt{R_0}\alpha^{-1}\|r(0)\| = 2\kappa\|r(0)\|.$$

Combining this with $\|r'(T) - \bar{r}(T)\| \leq \|r'(T)\| + \|\bar{r}(T)\| \leq 2\|r(0)\| = 2\sqrt{2R_0}$ implies (9). Thus, we have shown Theorem 3.2, which is the result of Theorem 1.2 if we replace $r$ by $r'$. The actual proof of the theorem handles the time-varying kernel $K_t$, and is in Appendix A.

### 3.2 Proof ideas for Theorem 1.3

The converse in Theorem 1.3 is achieved in the simple case where $h(w) = aw + \frac{1}{2}bw^2$ for $a = \frac{1}{\sqrt{T}}$ and $b = \text{Lip}(Dh)$, and $w_0 = 0$ and $R(y) = \frac{1}{2}(y - \sqrt{2R_0})^2$, as we show in Appendix B by direct calculation.

## 4 Discussion

A limitation of our result is that it applies only to the gradient flow, which corresponds to SGD with infinitesimally small step size. However, larger step sizes are beneficial for generalization in practice (see, e.g., Li et al. (2021); Andriushchenko et al. (2022)), so it would be interesting to understand the validity of the NTK approximation in that setting. Another limitation is that our result applies only to the square loss, and not to other popular losses such as the cross-entropy loss. Indeed, the known bounds in the setting of general losses require either a "well-conditioning" assumption, or taking $\alpha$ exponential in the training time $T$ (Chizat et al., 2019). Can one prove bounds of analogous to Theorem 1.2 for more general losses, with $\alpha$ depending polynomially on $T$, and without conditioning assumptions?

A natural question raised by our bounds in Theorems 1.2 and 1.3 is: how do the dynamics behave just outside the regime where the NTK approximation is valid? For models $h$ where $\text{Lip}(h)$ and $\text{Lip}(Dh)$ are bounded by a constant, can we understand the dynamics in the regime where $T \geq C\alpha$ for some large constant $C$ and $\alpha \gg C$, at the edge of the lazy training regime?

## Acknowledgements

We thank Emmanuel Abbe, Samy Bengio, and Joshua Susskind for stimulating and helpful discussions. EB also thanks Apple for the company's generous support through the AI/ML fellowship.

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

## A Proof of Theorem 1.2

### A.1 Notations

We let $R_0 := R(\alpha h(\boldsymbol{w}_0))$ denote the loss at initialization. We define the residuals $r(t), \bar{r}(t) \in \mathcal{F}$ under training the original model and the linearized model as

$$r(t) = y^* - \alpha h(\boldsymbol{w}(t)), \text{ and } \bar{r}(t) = y^* - \alpha \bar{h}(\bar{\boldsymbol{w}}(t)).$$

Since the evolution of $\boldsymbol{w}$ and $\bar{\boldsymbol{w}}$ is given by the gradient flow in (1) and (3), the residuals evolve as follows. We write $Dh^\top = \left(\frac{dh}{d\boldsymbol{w}}\right)^\top$ to denote the adjoint of $Dh = \frac{dh}{d\boldsymbol{w}}$.

$$\frac{dr}{dt} = \frac{dr}{d\boldsymbol{w}}\frac{d\boldsymbol{w}}{dt} = -\alpha Dh(\boldsymbol{w})(-\nabla F_\alpha(\boldsymbol{w})) = \alpha D(\boldsymbol{w})(\frac{1}{\alpha}Dh(\boldsymbol{w})^\top \nabla R(\alpha h(\boldsymbol{w}))) = -Dh(\boldsymbol{w})Dh(\boldsymbol{w})^\top r,$$

since $\nabla R(\alpha h(\boldsymbol{w})) = -(y^* - \alpha h(\boldsymbol{w})) = -r$. An analogous result can be derived for the residual $\bar{r}$ under the linearized dynamics:

$$\frac{d\bar{r}}{dt} = -D\bar{h}(\bar{\boldsymbol{w}})D\bar{h}(\bar{\boldsymbol{w}})^\top \bar{r}.$$

For any time $t \geq 0$, define the kernel $K_t : \mathcal{F} \to \mathcal{F}$ as

$$K_t := Dh(\boldsymbol{w}(t))Dh(\boldsymbol{w}(t))^\top.$$

Since $K_0 = Dh(\boldsymbol{w}(0))Dh(\boldsymbol{w}(0))^\top = D\bar{h}(\bar{\boldsymbol{w}}(t))D\bar{h}(\bar{\boldsymbol{w}}(t))^\top$, we can write the dynamics in compact form:

$$\frac{dr}{dt} = -K_t r \quad \text{and} \quad \frac{d\bar{r}}{dt} = -K_0 \bar{r}. \tag{10}$$

### A.2 Proof overview

Our proof of Theorem 1.2 is outlined in the flowchart in Figure 1. First, we use Lemma A.1 to argue that the change in the weights is bounded above during training. This implies several basic facts about the continuity and boundedness of the kernel $K_t$ over time, which are written as Lemmas A.2, A.3, A.4 A.5. These lemmas allow us to write the dynamics of $r$ and $\bar{r}$ using product integrals Dollard & Friedman (1984) in Lemma A.6. Product integrals are a standard tool in differential equations and quantum mechanics, but known results on product integrals do not suffice to prove our theorem. Thus, in Theorem A.8, we prove a new operator-norm bound between differences of product integrals. Theorem A.8 implies our main Theorem 1.2 when we specialize to the case of gradient-flow training of a model, and combine it with the weight-change bound from Lemma A.1. The bulk of the proof is dedicated to showing Theorem A.8. The proof is an interpolation argument, where we interpolate between the two product integrals being compared, and show that the operator norm of the derivative of the interpolation path is bounded. In order to show this, we require several new technical bounds – most notably Claim 6, which shows that for any matrices $X, Y \in \mathbb{R}^{n \times d}$ and any $t \geq 0$ we have

$$\|e^{-(X+Y)(X+Y)^\top t}(X+Y) - e^{-XX^\top t}X\| \leq 3\|Y\|.$$

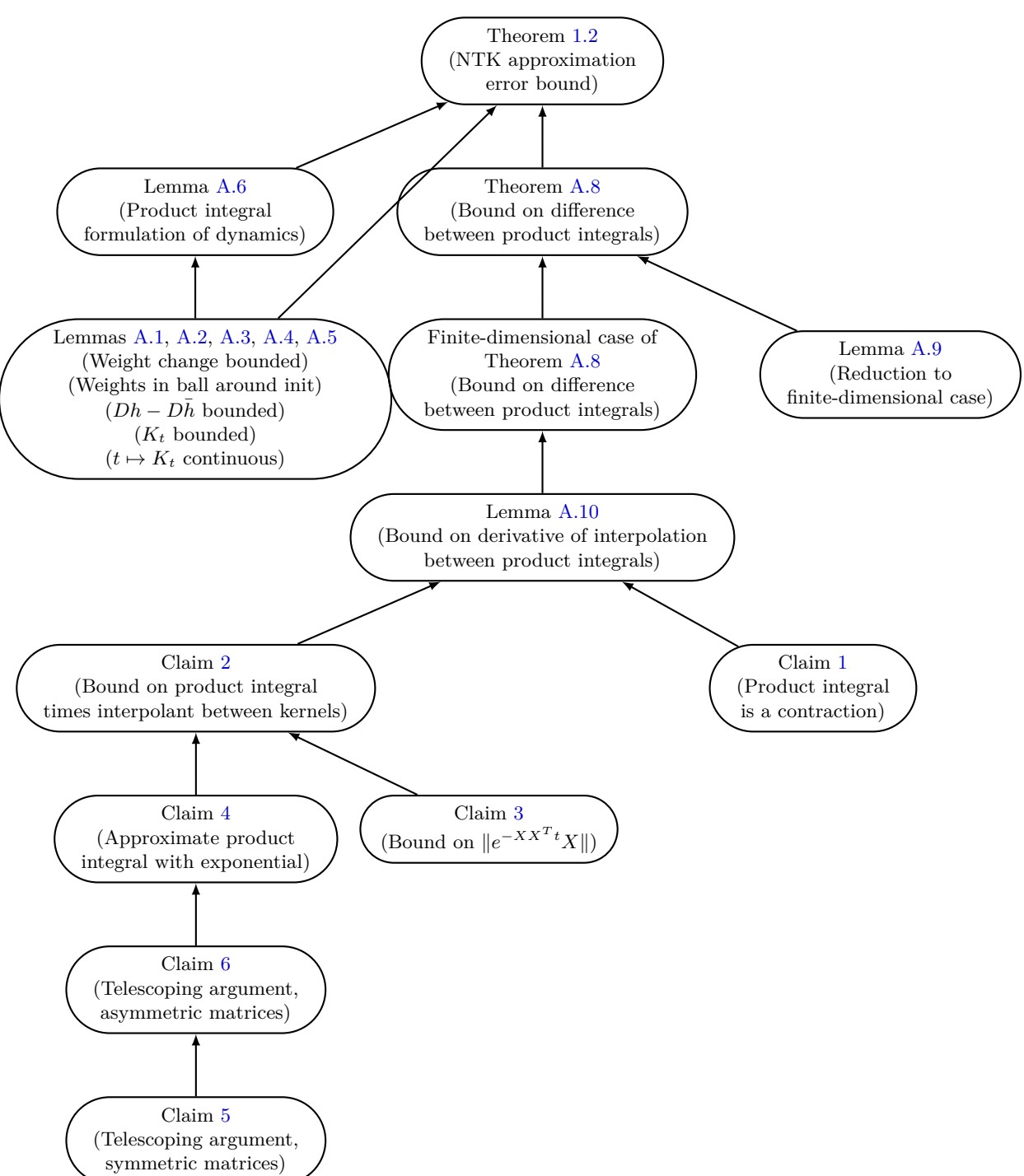

Figure 1: Proof structure.

### A.3 Basic facts about boundedness and continuity of the kernel

We begin with a series of simple propositions. Recall Proposition 3.1, which we restate below with a slight strengthening in (11) which will be needed later on.

**Lemma A.1** (Restatement of Proposition 3.1). *For any time $t$,*

$$\|\boldsymbol{w}(t) - \boldsymbol{w}_0\| \leq \sqrt{tR_0}/\alpha \quad and \quad \|\bar{\boldsymbol{w}}(t) - \boldsymbol{w}_0\| \leq \sqrt{tR_0}/\alpha\,.$$

*and furthermore*

$$\int_0^t \|\frac{d\boldsymbol{w}}{d\tau}\|d\tau \leq \sqrt{tR_0}/\alpha \quad and \quad \int_0^t \|\frac{d\bar{\boldsymbol{w}}}{d\tau}\|d\tau \leq \sqrt{tR_0}/\alpha\,. \tag{11}$$

*Proof.* The statement (11) is also implied by the proof of Proposition 3.1 in the main text. $\qquad\square$

The first implication is that the weights $\boldsymbol{w}(t)$ stay within the ball of radius $\rho$ around $\boldsymbol{w}_0$, during the time-span that we consider.

**Lemma A.2.** *For any time $0 \leq t \leq T$, we have $\|\boldsymbol{w}(t) - \boldsymbol{w}_0\| \leq \rho$.*

*Proof.* Immediate from Lemma A.1 and the fact that $T \leq \rho^2\alpha^2/R_0$. $\qquad\square$

This allows to use the bounds $\mathrm{Lip}(h)$ and $\mathrm{Lip}(Dh)$ on the Lipschitz constants of $h$ and $Dh$. Specifically, the kernels $K_t$ and $K_0$ stay close during training, in the sense that the difference between $Dh$ and $D\bar{h}$ during training is bounded.

**Lemma A.3.** *For any time $0 \leq t \leq T$, we have*

$$\|Dh(\boldsymbol{w}(t)) - D\bar{h}(\bar{\boldsymbol{w}}(t))\| \leq \mathrm{Lip}(Dh)\sqrt{tR_0}/\alpha$$

*Proof.* Since (a) $\bar{h}$ is the linearization of $h$ at $\boldsymbol{w}_0$, and (b) $\|\boldsymbol{w}(t) - \boldsymbol{w}_0\| \leq \min(\rho, \sqrt{tR_0}/\alpha)$ by Lemma A.2 and Lemma A.1,

$$\|Dh(\boldsymbol{w}(t)) - D\bar{h}(\bar{\boldsymbol{w}}(t))\| \overset{(a)}{=} \|Dh(\boldsymbol{w}(t)) - Dh(\boldsymbol{w}_0)\| \overset{(b)}{\leq} \mathrm{Lip}(Dh)\sqrt{tR_0}/\alpha\,.$$

$\qquad\square$

And therefore the kernel $K_t$ is bounded at all times $0 \leq t \leq T$ in operator norm.

**Lemma A.4.** *For any time $0 \leq t \leq T$, we have $\|K_t\| \leq 3\|Dh(\boldsymbol{w}(0))\|^2 + 2\mathrm{Lip}(Dh)tR_0/\alpha^2$.*

*Proof.* By triangle inequality and Lemma A.3,

$$\begin{aligned}
\|K_t - K_0\| &= \|Dh(\boldsymbol{w}(t))Dh(\boldsymbol{w}(t))^\top - Dh(\boldsymbol{w}(0))Dh(\boldsymbol{w}(0))^\top\| \\
&\leq \|Dh(\boldsymbol{w}(t))\|^2 + \|Dh(\boldsymbol{w}(0))\|^2 \\
&\leq (\|Dh(\boldsymbol{w}(t)) - Dh(\boldsymbol{w}(0))\| + \|Dh(\boldsymbol{w}(0))\|)^2 + \|Dh(\boldsymbol{w}(0))\|^2 \\
&\leq 3\|Dh(\boldsymbol{w}(0))\|^2 + 2\mathrm{Lip}(Dh)tR_0/\alpha^2\,.
\end{aligned}$$

$\qquad\square$

And finally we note that the kernel evolves continously in time.

**Lemma A.5.** *The map $t \mapsto K_t$ is continuous (in the operator norm topology) in the interval $[0, T]$.*

*Proof.* First, $t \mapsto \boldsymbol{w}(t)$ is continuous in time, since it solves the gradient flow. Second, we know that $\boldsymbol{w}(t)$ is in the ball of radius $\rho$ around $\boldsymbol{w}_0$ by Lemma A.2, and in this ball the map $\boldsymbol{w} \mapsto Dh(\boldsymbol{w})$ is continuous because $\mathrm{Lip}(Dh) < \infty$. Finally, $Dh \mapsto DhDh^\top$ is continuous. $\qquad\square$

## A.4 Product integral formulation of dynamics

Now we can present an equivalent formulation of the training dynamics (10) in terms of product integration. For any $0 \leq x \leq y \leq T$, let $P(y,x) : \mathbb{R}^p \to \mathcal{F}$ solve the operator integral equation

$$P(y,x) = I - \int_x^y K_t P(t,x) dt. \tag{12}$$

A solution $P(y,x)$ is guaranteed to exist and to be unique:

**Lemma A.6.** *The unique solution to the integral equation* (12) *is given as follows. For any* $0 \leq x \leq y \leq T$ *define* $s_{m,j} = (y-x)(j/m) + x$ *and* $\delta = (y-x)/m$, *and let* $P(y,x)$ *be the product integral*

$$P(y,x) := \prod_x^y e^{-K_s ds} := \lim_{m \to \infty} \prod_{j=1}^m e^{-\delta K_{s_j}} = \lim_{m \to \infty} e^{-\delta K_{s_m}} e^{-\delta K_{s_{m-1}}} \dots e^{-\delta K_{s_2}} e^{-\delta K_{s_1}}.$$

*Proof.* Existence, uniqueness, and the expression as an infinite product are guaranteed by Theorems 3.4.1, 3.4.2, and 3.5.1 of Dollard & Friedman (1984), since $t \mapsto K_t$ lies in $L_s^1(0,T)$, which is the space of "strongly integrable" functions on $[0,T]$ defined in Definition 3.3.1 of Dollard & Friedman (1984). This fact is guaranteed by the separability of $\mathcal{F}$ and the continuity and boundedness of $t \mapsto K_t$ (Lemmas A.4 and A.5). □

The operators $P(y,x)$ are the time-evolution operators corresponding to the differential equation (10) for the residual error $r_t$. Namely, for any time $0 \leq t \leq T$,

$$r_t = P(t,0)r_0.$$

On the other hand, the solution to the linearized dynamics (10) is given by

$$\bar{r}_t = e^{-K_0 t} r_0,$$

since $e^{-K_0 t}$ is the time-evolution operator when the kernel does not evolve with time.

## A.5 Error bound between product integrals

To prove Theorem 1.2, it suffices to bound $\|P(t,0) - e^{-K_0 t}\|$, the difference of the time-evolution operators under the full dynamics versus the linearized dynamics. We will do this via a general theorem. To state it, we must define the total variation norm of a time-indexed sequence of operators:

**Definition A.7.** Let $\{C_t\}_{t \in [x,y]}$ be a sequence of time-bounded operators $C_t : \mathbb{R}^p \to \mathcal{F}$ so that $t \mapsto C_t$ is continuous in the interval $[x,y]$. Then the total variation norm of $\{C_t\}_{t \in [x,y]}$ is

$$\mathcal{V}(\{C_t\}_{t \in [x,y]}) = \sup_{P \in \mathcal{P}} \sum_{i=1}^{n_P - 1} \|C_{t_i} - C_{t_{i-1}}\|,$$

where the supremum is taken over partitions $\mathcal{P} = \{P = \{x = t_1 \leq t_2 \leq \cdots \leq t_{n_P - 1} \leq t_{n_P} = y\}\}$ of the interval $[x,y]$.

We may now state the general result.

**Theorem A.8.** *Let* $\mathcal{F}$ *be a separable Hilbert space, and let* $\{A_t\}_{t \in [0,T]}, \{B_t\}_{t \in [0,T]}$ *be time-indexed sequences of bounded operators* $A_t : \mathbb{R}^p \to \mathcal{F}$ *and* $B_t : \mathbb{R}^p \to \mathcal{F}$ *such that* $t \mapsto A_t$ *and* $t \mapsto B_t$ *are continuous in* $[0,T]$. *Then,*

$$\| \prod_0^T e^{-A_s A_s^\top ds} - \prod_0^T e^{-B_s B_s^\top ds} \| \leq (\sup_{t \in [0,T]} \|A_t - B_t\|)(2\sqrt{T} + 3T \cdot \mathcal{V}(\{A_t - B_t\}_{t \in [0,T]})).$$

If we can establish Theorem A.8, then we may prove Theorem 1.2 as follows.

*Proof of Theorem 1.2.* for each $t \geq 0$ we choose the linear operators $A_t, B_t : \mathbb{R}^p \to \mathcal{F}$ by $A_t = Dh(\boldsymbol{w}(t))$ and $B_t = Dh(\boldsymbol{w}(0))$ so that $A_t A_t^\top = K_t$ and $B_t B_t^\top = K_0$. We know that $A_t, B_t$ are bounded by Lemma A.4, and that $t \mapsto A_t$ is continuous by Lemma A.5. (Also $t \mapsto B_t$ is trivially continuous). So we may apply Theorem A.8 to bound the difference in the residuals.

We first bound the total variation norm of $\{A_t - B_t\}_{t \in [0,T]}$, By (a) the fact from Lemma A.2 that $\boldsymbol{w}(t)$ is in a ball of radius at most $\rho$ around $\boldsymbol{w}_0$ where $\boldsymbol{w} \mapsto Dh(\boldsymbol{w})$ is $\mathrm{Lip}(Dh)$-Lipschitz; (b) the fact that $t \mapsto \boldsymbol{w}(t)$ is differentiable, since it solves a gradient flow; and (c) Lemma A.1,

$$
\begin{aligned}
\mathcal{V}(\{A_t - B_t\}_{t\in[0,T]}) &= \sup_{P \in \mathcal{P}} \sum_{i=1}^{n_P - 1} \|Dh(\boldsymbol{w}(t_{i+1})) - Dh(\boldsymbol{w}(0)) - Dh(\boldsymbol{w}(t_i)) + Dh(\boldsymbol{w}(0))\| \\
&= \sup_{P \in \mathcal{P}} \sum_{i=1}^{n_P - 1} \|Dh(\boldsymbol{w}(t_{i+1})) - Dh(\boldsymbol{w}(t_i))\| \\
&\overset{(a)}{\leq} \mathrm{Lip}(Dh) \sup_{P \in \mathcal{P}} \sum_{i=1}^{n_P - 1} \|\boldsymbol{w}(t_{i+1}) - \boldsymbol{w}(t_i)\| \\
&\overset{(b)}{=} \mathrm{Lip}(Dh) \int_0^T \|\frac{d\boldsymbol{w}}{dt}\| dt \\
&\overset{(c)}{\leq} \mathrm{Lip}(Dh)\sqrt{TR_0}/\alpha
\end{aligned}
$$

When we plug the above expression into Theorem A.8, along with the bound $\|A_t - B_t\| \leq \mathrm{Lip}(Dh)\sqrt{tR_0}/\alpha$ from Lemma A.3, we obtain:

$$
\begin{aligned}
\|r_T - \bar{r}_T\| = \|(\prod_{s=0}^{T} e^{-K_s ds} - \prod_{s=0}^{T} e^{-K_0 ds})r_0\| \\
\leq (\mathrm{Lip}(Dh)\sqrt{TR_0}/\alpha)(2\sqrt{T} + 3\mathrm{Lip}(Dh)T^{3/2}\sqrt{R_0}/\alpha)\sqrt{2R_0} \\
= (2\kappa + 3\kappa^2)\sqrt{2R_0},
\end{aligned}
$$

where $\kappa = \frac{T}{\alpha}\mathrm{Lip}(Dh)\sqrt{R_0}$.

Also note that

$$
\|r_T - \bar{r}_T\| \leq \|r_T\| + \|\bar{r}_T\| = \sqrt{2R(\alpha h(\boldsymbol{w}(T)))} + \sqrt{2R(\alpha \bar{h}(\bar{\boldsymbol{w}}(T)))} \leq 2\sqrt{2R_0},
$$

since the gradient flow does not increase the risk. So

$$
\|r_T - \bar{r}_T\| \leq \min(2\kappa + 3\kappa^2, 2)\sqrt{2R_0} \leq 6\kappa\sqrt{R_0} .
$$

$\square$

It remains only to prove Theorem A.8.

## A.6 Reduction to the finite-dimensional case

We first show that in order to prove Theorem A.8, it suffices to consider the case where $\mathcal{F}$ is a finite-dimensional Hilbert space. The argument is standard, and uses the fact that $\mathcal{F}$ has a countable orthonormal basis.

**Lemma A.9.** *Theorem A.8 is true for general separable Hilbert spaces $\mathcal{F}$ if it is true whenever $\mathcal{F}$ is finite-dimensional.*

*Proof.* Suppose that $\mathcal{F}$ is a separable Hilbert space and $A_t, B_t : \mathbb{R}^p \to \mathcal{F}$ satisfy the hypotheses of Theorem A.8. Let $\{f_i\}_{i \in \mathbb{N}}$ be a countable orthonormal basis for $\mathcal{F}$, which is guaranteed by separability of $\mathcal{F}$. For any $n$, let

$\mathcal{P}_n : \mathcal{F} \to \mathcal{F}$ be the linear projection operator defined by

$$\mathcal{P}_n(f_i) = \begin{cases} f_i, & 1 \le i \le n \\ 0, & \text{otherwise} \end{cases}.$$

By (a) Duhamel's formula in Theorem 3.5.8 of Dollard & Friedman (1984), (b) the fact that $\|e^{-A_s A_s^\top}\| \le 1$ and $\|e^{-\mathcal{P}_n A_s A_s^\top \mathcal{P}_n^\top}\| \le 1$ because $A_s A_s^\top$ and $\mathcal{P}_n A_s A_s^\top \mathcal{P}_n^\top$ are positive semidefinite.

$$\|\prod_0^T e^{-A_s A_s^\top ds} - \prod_0^T e^{-\mathcal{P}_n A_s A_s^\top \mathcal{P}_n^\top ds}\|$$

$$\stackrel{(a)}{=} \| \int_0^T \left( \prod_\tau^T e^{-A_s A_s^\top ds} \right) (A_\tau A_\tau^\top - \mathcal{P}_n A_\tau A_\tau^\top \mathcal{P}_n^\top) \left( \prod_0^\tau e^{-\mathcal{P}_n A_s A_s^\top \mathcal{P}_n^\top ds} \right) d\tau \|$$

$$\le \int_0^T \| \prod_\tau^T e^{-A_s A_s^\top ds}\| \| A_\tau A_\tau^\top - \mathcal{P}_n A_\tau A_\tau^\top \mathcal{P}_n^\top \| \| \prod_0^\tau e^{-\mathcal{P}_n A_s A_s^\top \mathcal{P}_n^\top ds} \| d\tau$$

$$\stackrel{(b)}{\le} \int_0^T \| A_\tau A_\tau^\top - \mathcal{P}_n A_\tau A_\tau^\top \mathcal{P}_n^\top \| d\tau \tag{13}$$

We have chosen $\mathcal{P}_n$ so that for any bounded linear operator $M : \mathcal{F} \to \mathcal{F}$, we have $\lim_{n \to \infty} \mathcal{P}_n M \mathcal{P}_n = M$. Since $\tau \mapsto A_\tau A_\tau^\top$ is continuous in $\tau$, and $A_\tau$ is bounded for each $\tau$, the expression in (13) converges to 0 as $n \to \infty$. By triangle inequality, we conclude that

$$\|\prod_0^T e^{-A_s A_s^\top ds} - \prod_0^T e^{-B_s B_s^\top ds}\| \le \limsup_{n \to \infty} \|\prod_0^T e^{-\mathcal{P}_n A_s A_s^\top \mathcal{P}_n^\top ds} - \prod_0^T e^{-\mathcal{P}_n B_s B_s^\top \mathcal{P}_n^\top ds}\|.$$

Notice that $\mathcal{P}_n A_t$ and $\mathcal{P}_n B_t$ are bounded maps from $\mathbb{R}^p$ to $\text{span}\{f_1, \ldots, f_n\}$, and $t \mapsto \mathcal{P}_n A_t$ and $t \mapsto \mathcal{P}_n B_t$ are continuous and bounded. So using the theorem in the case where $\mathcal{F}$ is finite-dimensional, the right-hand side can be bounded by

$$\limsup_{n \to \infty} \|\prod_0^T e^{-\mathcal{P}_n A_s A_s^\top \mathcal{P}_n^\top ds} - \prod_0^T e^{-\mathcal{P}_n B_s B_s^\top \mathcal{P}_n^\top ds}\|$$

$$\le \limsup_{n \to \infty} (\sup_{t \in [0,T]} \|\mathcal{P}_n A_t - \mathcal{P}_n B_t\|)(2\sqrt{T} + 3T \cdot \mathcal{V}(\{\mathcal{P}_n A_t - \mathcal{P}_n B_t\}_{t \in [0,T]})dt)$$

$$= (\sup_{t \in [0,T]} \|A_t - B_t\|)(2\sqrt{T} + 3T \cdot \mathcal{V}(\{A_t - B_t\}_{t \in [0,T]})dt).$$

$\square$

## A.7 Bound for the finite-dimensional case

We conclude by proving Theorem A.8 in the finite-dimensional case, where we write $A_t, B_t \in \mathbb{R}^{n \times d}$ as time-indexed matrices. In order to prove a bound, we will interpolate between the dynamics we wish to compare. Let

$$C_t = B_t - A_t \in \mathbb{R}^{n \times d}$$

For any $\zeta \in [0, 1]$ and $t \in [0, T]$, we define the kernel

$$K_{t,\zeta} = (A_t + \zeta C_t)(A_t + \zeta C_t)^\top \in \mathbb{R}^{n \times n}.$$

This interpolates between the kernels of interest, since on the one hand, $K_{t,0} = A_t A_t^\top$ and on the other $K_{t,1} = B_t B_t^\top$. For any $x, y \in [0, T]$, let

$$P(y, x; \zeta) = \prod_x^y e^{-K_{t,\zeta} dt} \in \mathbb{R}^{n \times n}.$$

The derivative $\frac{\partial}{\partial \zeta} P(y, x; \zeta)$ exists by Theorem 1.5.3 of Dollard & Friedman (1984), since (i) $K_{t,\zeta}$ is continuous in $t$ for each fixed $\zeta$, (ii) $K_{t,\zeta}$ is differentiable in $\zeta$ in the $L^1$ sense[4], and (iii) has a partial derivative $\frac{\partial K_{t,\zeta}}{\partial \zeta}$ that is integrable in $t$. The formula for $\frac{\partial}{\partial \zeta} P(y, x; \zeta)$ is given by the following formula, which generalizes the integral representation of the exponential map:[5]

$$\frac{\partial}{\partial \zeta} P(y, x; \zeta) = -\int_x^y P(y, t; \zeta) \frac{\partial K_{t,\zeta}}{\partial \zeta} P(t, x; \zeta) dt \tag{14}$$

Our main lemma is:

**Lemma A.10.** *For all $\zeta \in [0, 1]$, we have the bound*

$$\|\frac{\partial P(T, 0; \zeta)}{\partial \zeta}\| \leq (\sup_{t \in [0,T]} \|C_t\|)(2\sqrt{T} + 3T \cdot \mathcal{V}(\{C_t\}_{t \in [0,T]})).$$

This lemma suffices to prove Theorem A.8.

*Proof of Theorem A.8.* Using the fundamental theorem of calculus,

$$\|\prod_0^T e^{-A_t A_t^\top dt} - \prod_0^T e^{-B_t B_t^\top dt}\| = \|P(T, 0; 1) - P(T, 0; 0)\| \leq \int_0^1 \|\frac{\partial P(T, 0; \zeta)}{\partial \zeta}\| d\zeta,$$

which combined with Lemma A.10 proves Theorem A.8. □

## A.8 Proof of Lemma A.10

By a direct calculation,

$$\frac{\partial K_{t,\zeta}}{\partial \zeta} = (A_t + \zeta C_t) C_t^\top + C_t (A_t + \zeta C_t)^\top,$$

so, by (14),

$$\frac{\partial}{\partial \zeta} P(T, 0; \zeta) = -\int_0^T P(T, t; \zeta)((A_t + \zeta C_t) C_t^\top + C_t (A_t + \zeta C_t)^\top) P(t, 0; \zeta) dt$$

$$= \underbrace{-\int_0^T P(T, t; \zeta)(A_t + \zeta C_t) C_t^\top P(t, 0; \zeta) dt}_{M_1} \underbrace{-\int_0^T P(T, t; \zeta) C_t (A_t + \zeta C_t)^\top P(t, 0; \zeta) dt}_{M_2}.$$

The arguments are similar for bounding $M_1$ and $M_2$, so we only bound $M_1$. We will need two technical bounds, whose proofs are deferred to Section A.9.

**Claim 1.** *For any $0 \leq t \leq T$, we have $\|P(t, 0; \zeta)\| \leq 1$.*

**Claim 2.** *For any $0 \leq t \leq T$, we have $\|P(T, t; \zeta)(A_t + \zeta C_t)\| \leq \frac{1}{2\sqrt{T-t}} + 3\mathcal{V}(\{C_s\}_{s \in [t,T]})$.*

Using (a) Claim 1, and (b) Claim 2,

$$\|M_1\| \overset{(a)}{\leq} (\sup_{t \in [0,T]} \|C_t\|) \int_0^T \|P(T, t; \zeta)(A_t + \zeta C_t)\| dt \tag{15}$$

$$\overset{(b)}{\leq} (\sup_{t \in [0,T]} \|C_t\|) \left(\int_0^T \frac{1}{2\sqrt{T-t}} + 3\mathcal{V}(\{C_s\}_{s \in [t,T]}) dt\right) \tag{16}$$

$$= (\sup_{t \in [0,T]} \|C_t\|)(\sqrt{T} + 3\int_0^T \mathcal{V}(\{C_s\}_{s \in [t,T]}) dt). \tag{17}$$

---

[4]For any $\zeta \in [0, 1]$, $\lim_{\zeta' \to \zeta} \int_0^T \|\frac{K_{t,\zeta'} - K_{t,\zeta}}{\zeta' - \zeta} - \frac{\partial K_{t,\zeta}}{\partial \zeta}\| dt = 0$, since the matrices $A_t, B_t$ are uniformly bounded.

[5]The proof is a consequence of Duhamel's formula. This a tool used in a variety of contexts, including perturbative analysis of path integrals in quantum mechanics Dollard & Friedman (1984).

Lemma A.10 is proved by noting that, symmetrically,

$$\|M_2\| \le (\sup_{t \in [0,T]} \|C_t\|)(\sqrt{T} + 3\int_0^T \mathcal{V}(\{C_s\}_{s \in [0,t]})dt) \,,$$

and, for any $t \in [0, T]$,

$$\mathcal{V}(\{C_s\}_{s \in [0,t]}) + \mathcal{V}(\{C_s\}_{s \in [t,T]}) = \mathcal{V}(\{C_s\}_{s \in [0,T]}) \,.$$

### A.9 Deferred proofs of Claims 1 and 2

**Claim 1.** *For any $0 \le t \le T$, we have $\|P(t, 0; \zeta)\| \le 1$.*

*Proof.* This follows from the definition of the product integral as an infinite product, and the fact that each term $e^{-\delta K_{t_i, \zeta}}$ in the product has norm at most 1 because $K_{t_i, \zeta}$ is positive semidefinite. $\square$

In order to prove Claim 2, we need two more claims:

**Claim 3.** *For any $X \in \mathbb{R}^{n \times d}$ and $t \ge 0$,*

$$\|e^{-XX^\top t}X\| \le \frac{1}{2\sqrt{t}}$$

*Proof.* Since $XX^\top$ is positive semidefinite,

$$\|e^{-XX^\top}X\| = \sqrt{\|e^{-XX^\top t}XX^\top e^{-XX^\top t}\|} \le \sup_{\lambda \ge 0}\sqrt{e^{-\lambda t}\lambda e^{-\lambda t}} = \sup_{\lambda \ge 0} e^{-\lambda}\sqrt{\lambda/t} \le \frac{1}{2\sqrt{t}}.$$

$\square$

**Claim 4.** *For any time-indexed sequence of matrices $(X_t)_{t \in [0,T]}$ in $\mathbb{R}^{n \times d}$ such that $t \mapsto X_t$ is continuous in $[0, T]$, and any $0 \le a \le b \le T$,*

$$\left\|e^{-X_b X_b^\top (b-a)}X_b - \left(\prod_a^b e^{-X_s X_s^\top ds}\right)X_a\right\| \le 3\mathcal{V}(\{X_s\}_{s \in [a,b]})$$

This latter claim shows that we can approximate the product integral with an exponential. The proof is involved, and is provided in Section A.10. Assuming the previous two claims, we may prove Claim 2.

**Claim 2.** *For any $0 \le t \le T$, we have $\|P(T, t; \zeta)(A_t + \zeta C_t)\| \le \frac{1}{2\sqrt{T-t}} + 3\mathcal{V}(\{C_s\}_{s \in [t,T]})$.*

*Proof.* By Claim 3, since $K_{T, \zeta} = (A_T + \zeta C_T)(A_T + \zeta C_T)^\top$,

$$\|e^{-K_{T,\zeta}(T-t)}(A_T + \zeta C_T)\| \le \frac{1}{2\sqrt{T-t}} \,.$$

By the triangle inequality, it remains to prove

$$\|e^{-K_{T,\zeta}(T-t)}(A_T + \zeta C_T) - P(T, t; \zeta)(A_t + \zeta C_t)^\top\| \le 3\mathcal{V}(\{C_s\}_{s \in [t,T]}),$$

and this is implied by Claim 4, defining $X_t = A_t + \zeta C_t$ and $a = t$, $b = T$. $\square$

## A.10 Deferred proof of Claim 4

The proof will be by interpolation, using the integral representation of the exponential map provided in (14), similarly to the main body of the proof, but of course interpolating with respect to a different parameter. We begin the following claim, which we will subsequently strengthen.

**Claim 5.** *For any symmetric $X, Y \in \mathbb{R}^{n \times n}$ and $t \geq 0$,*

$$\|e^{-(X+Y)^2 t}(X+Y) - e^{-X^2 t} X\| \leq 3\|Y\|$$

*Proof.* For any $\tau \in [0,1]$, define $X(\tau) = X + \tau Y$, which interpolates between $X$ and $Y$. Then by (a) the derivative of the exponential map in (14), and (b) the fact that $X'(\tau) = Y$ and $\|e^{-X^2(\tau)t/2}\| \leq 1$,

$$\|e^{-(X+Y)^2 t}(X+Y) - e^{-X^2 t} X\|$$

$$= \|\int_0^1 \frac{d}{d\tau}\left(e^{-X^2(\tau)t} X(\tau)\right) d\tau\|$$

$$= \|\int_0^1 \frac{d}{d\tau}\left(e^{-X^2(\tau)t/2} X(\tau) e^{-X^2(\tau)t/2}\right) d\tau\|$$

$$\leq \sup_{\tau \in [0,1]} \|\frac{d}{d\tau} e^{-X^2(\tau)t/2} X(\tau) e^{-X^2(\tau)t/2}\|$$

$$\overset{(a)}{=} \sup_{\tau \in [0,1]} \|e^{-X^2(\tau)t/2} X'(\tau) e^{-X^2(\tau)t/2}$$

$$- (t/2) \int_0^1 e^{-(1-s)X^2(\tau)t/2}(X'(\tau)X(\tau) + X(\tau)X'(\tau)) e^{-sX^2(\tau)t/2} X(\tau) e^{-X^2(\tau)t/2} ds$$

$$- (t/2) \int_0^1 e^{-X^2(\tau)t/2} X(\tau) e^{-(1-s)X^2(\tau)t/2}(X'(\tau)X(\tau) + X(\tau)X'(\tau)) e^{-sX^2(\tau)t/2} ds\|$$

$$\overset{(b)}{\leq} \sup_{\tau \in [0,1]} \|Y\| + \underbrace{\|(t/2) \int_0^1 e^{-(1-s)X^2(\tau)t/2}(YX(\tau) + X(\tau)Y) e^{-sX^2(\tau)t/2} X(\tau) e^{-X^2(\tau)t/2} ds\|}_{T_1}$$

$$+ \underbrace{\|(t/2) \int_0^1 e^{-X^2(\tau)t/2} X(\tau) e^{-(1-s)X^2(\tau)t/2}(YX(\tau) + X(\tau)Y) e^{-sX^2(\tau)t/2} ds\|}_{T_2},$$

and by (a) $\sup_{\lambda \geq 0} \lambda e^{-\lambda^2 t/2} = 1/\sqrt{et}$, and (b) $\|e^{-M}\| \leq 1$ if $M$ is p.s.d.,

$$T_1 \leq (t/2) \int_0^1 \|e^{-(1-s)X^2(\tau)t/2}(YX(\tau) + X(\tau)Y) e^{-sX^2(\tau)t/2}\| \|X(\tau) e^{-X^2(\tau)t/2}\| ds$$

$$\overset{(a)}{\leq} (t/2) \int_0^1 \|e^{-(1-s)X^2(\tau)t/2}(YX(\tau) + X(\tau)Y) e^{-sX^2(\tau)t/2}\| \frac{1}{\sqrt{et}} ds$$

$$\leq \frac{\sqrt{t}}{2\sqrt{e}} \int_0^1 \|e^{-(1-s)X^2(\tau)t/2}\| \|Y\| \|X(\tau) e^{-sX^2(\tau)t/2}\| + \|e^{-(1-s)X^2(\tau)t/2} X(\tau)\| \|Y\| \|e^{-sX^2(\tau)t/2}\| ds$$

$$\overset{(a)}{\leq} \frac{\sqrt{t}}{2e} \int_0^1 \|e^{-(1-s)X^2(\tau)t/2}\| \|Y\| \frac{1}{\sqrt{st}} + \frac{1}{\sqrt{(1-s)t}} \|Y\| \|e^{-sX^2(\tau)t/2}\| ds$$

$$\overset{(b)}{\leq} \frac{\|Y\|}{2e} \int_0^1 \frac{1}{\sqrt{s}} + \frac{1}{\sqrt{(1-s)}} ds$$

$$= \frac{\|Y\|}{2e}(2\sqrt{s} - 2\sqrt{1-s})_0^1$$

$$\leq \|Y\|.$$

Similarly, $T_2 \leq \|Y\|$. $\qquad \square$

**Claim 6.** *For any $X, Y \in \mathbb{R}^{n \times d}$ (not necessarily symmetric) and $t \geq 0$,*

$$\|e^{-(X+Y)(X+Y)^\top t}(X+Y) - e^{-XX^\top t}X\| \leq 3\|Y\|.$$

*Proof.* We will use Claim 5, combined with a general method for lifting facts from symmetric matrices to asymmetric matrices (see, e.g., Tao (2011) for other similar arguments). Assume without loss of generality that $n = d$, since otherwise we can pad with zeros. Define $\bar{n} = 2n$ and

$$\bar{X} = \begin{bmatrix} 0 & X^\top \\ X & 0 \end{bmatrix} \in \mathbb{R}^{\bar{n} \times \bar{n}} \text{ and } \bar{Y} = \begin{bmatrix} 0 & Y^\top \\ Y & 0 \end{bmatrix} \in \mathbb{R}^{\bar{n} \times \bar{n}}.$$

These are symmetric matrices by construction. Furthermore,

$$\bar{X}^2 = \begin{bmatrix} X^\top X & 0 \\ 0 & XX^\top \end{bmatrix} \text{ and } (\bar{X} + \bar{Y})^2 = \begin{bmatrix} (X+Y)^\top(X+Y) & 0 \\ 0 & (X+Y)(X+Y)^\top \end{bmatrix}.$$

Because of the block-diagonal structure of these matrices,

$$e^{-\bar{X}^2 t} = \begin{bmatrix} e^{-X^\top X t} & 0 \\ 0 & e^{-XX^\top t} \end{bmatrix} \text{ and } e^{-(\bar{X}+\bar{Y})^2 t} = \begin{bmatrix} e^{-(X+Y)^\top(X+Y)t} & 0 \\ 0 & e^{-(X+Y)(X+Y)^\top t} \end{bmatrix}.$$

So

$$e^{-\bar{X}^2 t}\bar{X} = \begin{bmatrix} 0 & e^{-X^\top X t}X^\top \\ e^{-XX^\top t}X & 0 \end{bmatrix}.$$

Similarly,

$$e^{-(\bar{X}+\bar{Y})^2 t}(\bar{X}+\bar{Y}) = \begin{bmatrix} 0 & e^{-(X+Y)^\top(X+Y)t}(X+Y)^\top \\ e^{-(X+Y)(X+Y)^\top t}(X+Y) & 0 \end{bmatrix}.$$

For any matrix $M \in \mathbb{R}^{n \times n}$, we have $\|M\| = \sup_{\boldsymbol{v} \in \mathbb{S}^{n-1}} \|M\boldsymbol{v}\|$. So for any matrices $M_1, M_2 \in \mathbb{R}^{n \times n}$, we have

$$\left\| \begin{bmatrix} 0 & M_1 \\ M_2 & 0 \end{bmatrix} \right\| \geq \sup_{\boldsymbol{v} \in \mathbb{S}^{n-1}} \left\| \begin{bmatrix} 0 & M_1 \\ M_2 & 0 \end{bmatrix} \begin{bmatrix} \boldsymbol{v} \\ 0 \end{bmatrix} \right\| = \sup_{\boldsymbol{v} \in \mathbb{S}^{n-1}} \|M_2\boldsymbol{v}\| = \|M_2\|.$$

This means that, using Claim 5,

$$\|e^{-(X+Y)(X+Y)^\top t}(X+Y) - e^{-XX^\top t}X\| \leq \|e^{-(\bar{X}+\bar{Y})^2 t}(\bar{X}+\bar{Y}) - e^{\bar{X}^2 t}\bar{X}\| \leq 3\|\bar{Y}\|.$$

Finally, using the symmetry of $\bar{Y}$ and the block-diagonal structure of $\bar{Y}^2$,

$$\|\bar{Y}\| = \|\bar{Y}^2\|^{1/2} = \left\| \begin{bmatrix} Y^\top Y & 0 \\ 0 & YY^\top \end{bmatrix} \right\|^{1/2} = \max(\|Y^\top Y\|^{1/2}, \|YY^\top\|^{1/2}) = \|Y\|.$$

$\square$

We conclude by using these results to prove Claim 4 with a telescoping argument.

**Claim 4.** *For any time-indexed sequence of matrices $(X_t)_{t \in [0,T]}$ in $\mathbb{R}^{n \times d}$ such that $t \mapsto X_t$ is continuous in $[0, T]$, and any $0 \leq a \leq b \leq T$,*

$$\left\| e^{-X_b X_b^\top (b-a)} X_b - \left( \prod_a^b e^{-X_s X_s^\top ds} \right) X_a \right\| \leq 3\mathcal{V}(\{X_s\}_{s \in [a,b]})$$

*Proof.* We will do this by approximating the product integral by a finite product of $m$ matrices, and taking the limit $m \to \infty$. For any $m \geq 0$ and $j \in \{0, \ldots, m\}$, and $t \in [0, T]$, let $t_{m,j} = (b-a)(j/m) + a$, and define the finite-product approximation to the product integral for any $k \in \{0, \ldots, m\}$

$$P_{m,k} = \left( \prod_{j=k+1}^{m} Q_{m,j} \right) Q_{m,k}^k ,$$

where

$$Q_{m,j} = e^{-X_{t_{m,j}} X_{t_{m,j}}^\top (b-a)/m} .$$

Notice that for $k = 0$ we have

$$P_{m,0} = \left( \prod_{j=1}^{m} e^{-X_{t_{m,j}} X_{t_{m,j}}^\top (b-a)/m} \right) ,$$

and so by continuity of $s \mapsto X_s$ for $s \in [0, T]$ and boundedness of $X_s$, we know from Theorem 1.1 of Dollard & Friedman (1984) that

$$\lim_{m \to \infty} P_{m,0} = \prod_{a}^{b} e^{-X_s X_s^\top ds} .$$

This means that

$$\|e^{-X_b X_b^\top (b-a)} X_b - \left( \prod_a^b e^{-X_s X_s^\top ds} \right) X_a\| = \|P_{m,m} X_{t_{m,m}} - \left( \prod_a^b e^{-X_s X_s^\top ds} \right) X_a\|$$

$$= \lim_{m \to \infty} \|P_{m,m} X_{t_{m,m}} - P_{m,0} X_{t_{m,0}}\| .$$

Finally, telescoping and (a) using $\|Q_{m,j}\| \leq 1$ for all $j$, and (b) Claim 6,

$$\|P_{m,m} X_{t_{m,m}} - P_{m,1} X_{t_{m,1}}\| \leq \sum_{k=1}^{m} \|P_{m,k} X_{t_{m,k}} - P_{m,k-1} X_{t_{m,k-1}}\|$$

$$= \sum_{k=1}^{m} \| \left( \prod_{j=k+1}^{m} Q_{m,j} \right) (Q_{m,k}^k X_{t_{m,k}} - Q_{m,k}(Q_{m,k-1})^{k-1} X_{t_{m,k-1}}) \|$$

$$\stackrel{(a)}{\leq} \sum_{k=1}^{m} \|(Q_{m,k})^{k-1} X_{t_{m,k}} - (Q_{m,k-1})^{k-1} X_{t_{m,k-1}}\|$$

$$\stackrel{(b)}{\leq} \sum_{k=1}^{m} 3 \|X_{t_{m,k}} - X_{t_{m,k-1}}\|$$

$$\leq 3\mathcal{V}(\{X_s\}_{s \in [a,b]}).$$

The lemma follows by taking $m \to \infty$, since the bound is independent of $m$. $\qquad\square$

## B   Proof of Theorem 1.3

The converse bound in Theorem 1.2 is achieved in the simple case where $h : \mathbb{R} \to \mathbb{R}$ is given by $h(w) = aw + \frac{1}{2}bw^2$ for $a = \frac{1}{\sqrt{T}}$ and $b = \text{Lip}(Dh)$. We also let $w_0 = 0$ and $y^* = \sqrt{2R_0}$ so that all conditions are satisfied. The evolution of the residuals $r(t) = y^* - \alpha h(w(t))$ and $\bar{r}(t) = y^* - \alpha \bar{h}(\bar{w}(t))$ is given by

$$\frac{dr}{dt} = -K_t r \quad \text{and} \quad \frac{d\bar{r}}{dt} = -K_0 \bar{r},$$

where $K_t = Dh(w(t))Dh(w(t))^\top = (a + bw(t))^2$ and $K_0 = a^2$. Since $r = y^* - \alpha(aw + \frac{b}{2}w^2)$, we can express the evolution of the residuals $r$ and $\bar{r}$ as:

$$\frac{dr}{dt} = -(a^2 + 2b(y^* - r)/\alpha)r \quad \text{and} \quad \frac{d\bar{r}}{dt} = -a^2 \bar{r} . \tag{18}$$

Since $b(y^* - r)/\alpha \geq 0$ at all times, we must have $a^2 + 2b(y^* - r)/\alpha \geq a^2$. This means that at all times

$$r(t) \leq \bar{r}(t) = e^{-a^2 t} y^*.$$

So, at any time $t \geq T/2$,

$$r(t) \leq r(T/2) \leq \bar{r}(T/2) = e^{-(1/\sqrt{T})^2(T/2)} y^* = e^{-1/2} y^*,.$$

By plugging this into (18), for times $t \geq T/2$,

$$\frac{dr}{dt} \leq -(a^2 + 2\mathrm{Lip}(Dh)(1 - e^{-1/2})\sqrt{2R_0}/\alpha)r \leq -(a^2 + 1.1\kappa/T)r.$$

So, at time $T$, assuming that $\kappa \leq 1$ without loss of generality,

$$r(T) \leq r(T/2)e^{-(1/T + 1.1\kappa/T)(T/2)} = r(T/2)e^{-1/2 - 0.55\kappa} \leq y^* e^{-1} e^{-0.55\kappa} \leq y^* e^{-1}(1 - 0.4\kappa).$$

So

$$|\alpha h(w(T)) - \alpha \bar{h}(\bar{w}(T))| = |r(T) - \bar{r}(T)| \geq |e^{-1} y^* - (1 - 0.4\kappa)e^{-1} y^*| \geq 0.4\kappa e^{-1}\sqrt{2R_0} \geq \sqrt{R_0}/5.$$

$\square$

## C    Deferred details from Section 2

The bound on $\mathrm{Lip}(Dh)$ for 2-layer networks is below.

**Lemma C.1** (Bound on $\mathrm{Lip}(Dh)$ for mean-field 2-layer network)**.** *Suppose that there is a constant $K$ such that (i) the activation function $\sigma$ is bounded and has bounded derivatives $\|\sigma\|_\infty, \|\sigma'\|_\infty, \|\sigma''\|_\infty, \|\sigma'''\|_\infty \leq K$, (ii) the weights have bounded norm $\|\boldsymbol{a}\| + \|\boldsymbol{U}\| \leq K$, and (iii) the data points have bounded norm $\max_i \|\boldsymbol{x}_i\| \leq K$. Then there is a constant $K'$ depending only $K$ such that*

$$\mathrm{Lip}(Dh) \leq K'.$$

*Proof.* Let $p = m + md$ be the number of parameters of the network. Then $Dh \in \mathbb{R}^{n \times p}$ is

$$Dh = \frac{1}{\sqrt{n}} \begin{bmatrix} Df_{\boldsymbol{w}}(\boldsymbol{x}_1) \\ \vdots \\ Df_{\boldsymbol{w}}(\boldsymbol{x}_n) \end{bmatrix}.$$

So

$$\mathrm{Lip}(Dh) \leq \max_{i \in [n]} \mathrm{Lip}(Df_{\boldsymbol{w}}(\boldsymbol{x}_i)).$$

So for the 2-layer network,

$$
\begin{aligned}
\mathrm{Lip}(Dh) \leq{} & \max_{i \in [n]} \frac{1}{m} \mathrm{Lip}\big(\big[\sigma(\sqrt{m}\langle \boldsymbol{u}_1, \boldsymbol{x}_i\rangle) \quad \ldots \quad \sigma(\sqrt{m}\langle \boldsymbol{u}_m, \boldsymbol{x}_i\rangle)\big]\big) \\
& + \frac{1}{\sqrt{m}} \mathrm{Lip}\big(\big[a_1 \sigma'(\sqrt{m}\langle \boldsymbol{u}_1, \boldsymbol{x}_i\rangle)\boldsymbol{x}_i^\top \quad \ldots \quad a_m \sigma'(\sqrt{m}\langle \boldsymbol{u}_m, \boldsymbol{x}_i\rangle)\boldsymbol{x}_i^\top\big]\big) \\
={} & \max_{i \in [n]} \frac{1}{\sqrt{m}} \left\| \begin{bmatrix} \sigma'(\sqrt{m}\langle \boldsymbol{u}_1, \boldsymbol{x}_i\rangle)\boldsymbol{x}_i^\top & \boldsymbol{0} & \boldsymbol{0} & \ldots & \boldsymbol{0} \\ \boldsymbol{0} & \sigma'(\sqrt{m}\langle \boldsymbol{u}_2, \boldsymbol{x}_i\rangle)\boldsymbol{x}_i^\top & \boldsymbol{0} & \ldots & \boldsymbol{0} \\ \vdots & & & & \\ \boldsymbol{0} & & \ldots & \boldsymbol{0} & \sigma'(\sqrt{m}\langle \boldsymbol{u}_m, \boldsymbol{x}_i\rangle)\boldsymbol{x}_i^\top \end{bmatrix} \right\| \\
& + \frac{\|\boldsymbol{x}_i\|}{\sqrt{m}} \mathrm{Lip}\big(\big[a_1 \sigma'(\sqrt{m}\langle \boldsymbol{u}_1, \boldsymbol{x}_i\rangle) \quad \ldots \quad a_m \sigma'(\sqrt{m}\langle \boldsymbol{u}_m, \boldsymbol{x}_i\rangle)\big]\big) \\
\leq{} & \frac{\|\sigma'\|_\infty \|\boldsymbol{x}_i\|}{\sqrt{m}} + \frac{\|\boldsymbol{x}_i\|}{\sqrt{m}} \big\| \mathrm{diag}\big(\big[\sigma'(\sqrt{m}\langle \boldsymbol{u}_1, \boldsymbol{x}_i\rangle) \quad \ldots \quad \sigma'(\sqrt{m}\langle \boldsymbol{u}_m, \boldsymbol{x}_i\rangle)\big]\big) \big\| \\
& + \|\boldsymbol{x}_i\| \left\| \begin{bmatrix} a_1 \sigma''(\sqrt{m}\langle \boldsymbol{u}_1, \boldsymbol{x}_i\rangle)\boldsymbol{x}_i^\top & \boldsymbol{0} & \ldots & & \boldsymbol{0} \\ \vdots & & & & \vdots \\ \boldsymbol{0} & & \ldots & \boldsymbol{0} & a_m \sigma''(\sqrt{m}\langle \boldsymbol{u}_m, \boldsymbol{x}_i\rangle)\boldsymbol{x}_i^\top \end{bmatrix} \right\| \\
\leq{} & 2\frac{\|\sigma'\|_\infty \|\boldsymbol{x}_i\|}{\sqrt{m}} + \|\sigma''\|_\infty \|\boldsymbol{x}_i\|^2 \|\boldsymbol{a}\|_\infty
\end{aligned}
$$

$\square$

