# OpenReview forum: "Tight conditions for when the NTK approximation is valid"
_TMLR — Accepted by TMLR_

### Review · Reviewer_jka3 · 2023-07-05

**Summary Of Contributions:**

The authors tightened an NTK approximation result of Chizat and Bach (2019) based on time rescaling, and showed this result is tight with a matching lower bound.

I believe this submission exactly fits the purpose of TMLR: to publish correct and useful technical results, without judging for the result's "impact" or other subjective criteria. I would recommend accepting this submission as it is.

**Audience:**

Yes

**Claims And Evidence:**

Yes

**Requested Changes:**

N/A

**Strengths And Weaknesses:**

Strengths

The result is a clear improvement over existing work and is shown to have a matching lower bound, thereby providing a very satisfying conclusion to this problem.

In fact, there is very little to cherry pick, as the contributions are quite clear and easy to understand.

---

### Review · Reviewer_7ztJ · 2023-07-22

**Summary Of Contributions:**

The paper introduces a tighter analysis of the NTK evolution in the
most general setting introduced by Chizat et al. (2019).
In particular, their new bound holds for a much longer time horizon
($\alpha^2\rho^2$ vs $\alpha\rho/\text{Lip}(Dh)$) and is tighter ($\kappa$ vs $\kappa T\text{Lip}(Dh)$). Framing this in terms of initial rescaling and a fixed error budget epsilon, the time horizon is squared from Chizat et al.'s $O(\sqrt{\alpha\varepsilon})$ to $O(\alpha\varepsilon)$.

To achieve this the authors propose a new proof blueprint based on a product integral formulation of the dynamics of the residual, leveraging in a few points results from Chizat et al.

The main technical challange seem to be showing continuity and operator-boundedness guarantee for the main quantities involved in the process, which in turns makes it possible to leverage classical results from Dollard & Friedman (1984) for the evolution of bounded continuous operators. Once the main reformulation is achieved, the rest of the proof is a more straightforward (but still highly technical) sequence of bounding to recover a quantity comparable to Chizat et al.

Once they establish their tighter bound, the authors also provide a worst case example that achieves it, which prove that the bound is indeed tight under this specific set of assumptions.

**Audience:**

Yes

**Broader Impact Concerns:**

N/A as this is a theory paper

**Claims And Evidence:**

Yes

**Requested Changes:**

Add a short discussion of the necessity and impact of the assumptions, and a clarification of the $\sup_t$ to $K_T$ bound.

Beyond that, I think the paper as is achieves overall the threshold for acceptance in TMLR. However a broader discussion of the assumptions, more structured presentation of the proof, and an illustrative empirical validation would strengthen the contribution.

Minor:
- $\kappa$ is defined as part of Prop. 1.1 while it is a shared quantity used across theorems and should be independently defined as $\alpha, R_0$ and other common quantities are
- adding a proof flowchart/graph would help legibility of the appendix a lot.

**Strengths And Weaknesses:**

The strong points of the paper are:
- Clear improvement over current SOTA (Chizat et al.)
- General and novel approach that can probably be further refined/extended
- Thorough derivation, missing only a good recap (ideally a proof flowchart) for the extensive appendix
- Corresponding lower bound

The weaknesses:
- The result only holds for continuous flow, and is tight only under very general assumptions while real-world data might satisfy stronger assumptions
- At the same time the authors do not discuss how realistic the assumptions they inherit from Chizat et al. are. This is particularly important when these assumptions are playing an important role in the proof (e.g. assuming that $Dh$ is $\text{Lip}(Dh)$-Lipschitz in a ball of radius $\rho$ makes it much easier to prove boundedness of the operator)
- The proof sketch in Sec. 2 gives an intuition of the main result by saying that the $K_T$ process bounds any of the $K_t$ intermediate processes. However in the appendix the proof mainly relies on a $\sup_{t \in [0,T]}$ analysis, and while I find it plausible that that $K_T$ is a bound on this sup I cannot find a clear point where it is shown and the derivation is not immediate.

Minor:
- This is a theory paper so empirical validations are not strictly necessary. However the result would be strengthened by showing that it capture the real-world behaviour of NN optimization (the original goal of NTK). For example the authors could show empirically how close (or not, for "easy" data) real-world flows are to their bounds, how much finite steps make a real-world system deviate from their analysis. Finally, since the authors provide a constructive lower bound, it would have been nice to provide some real-world simulation to see if the designed model keeps close to the theoretical behaviour under finite steps.
- Section 2 is slightly confusing w.r.t. how novel the proof blueprint is. The authors start with Chizat et al.'s algebra-based proof, then introduce an intermediate algebra-based lemma, then switch to their integral-representation based proof but eventually return to the lemma. The flow can be greatly improved by clarifying which parts of the approach are novel and which are off-the-shelf.

---

> ### Author Response · Authors · 2023-08-18
> **Response to questions we did not answer in the general comment**
>
>
> Thanks for your review. We have addressed your other questions in the comment to all reviewers.
>
> * Regarding your question about $\sup_t$ analysis: in the main text we prove a simplified variant of the main theorem. We make this clearer in the revision. This simplified variant is close to the main theorem only if you accept the intuition that the $K_T$ kernel is the worst-case kernel for the $r$ and $\bar{r}$ residual evolutions to differ. The proof of the actual theorem in the appendix does not make such an assumption and therefore is more involved. The proof in the appendix holds for any $T$ such that $0 \leq T \leq \alpha^2 \rho^2 / R_0$, so in particular it also holds for any time $0 \leq t \leq T$.
>
> * We did not have time before the response deadline to add an experiment as you optionally suggested, but we might add one in a final version if we have time.

---

### Review · Reviewer_t2p5 · 2023-08-04

**Summary Of Contributions:**

This paper establishes a tighter bound on the condition required for NTK approximation.

**Audience:**

Yes

**Broader Impact Concerns:**

No concern.

**Claims And Evidence:**

Yes

**Requested Changes:**

See the weakness section.

**Strengths And Weaknesses:**

Strengths:

* In terms of the results, this paper shows that the NTK approximation can be achieved by setting $\alpha=O(T)$, which improves the previous result that requires $\alpha=O(T^2)$.

* This paper develops a more refined proof technique, which can be used in other theoretical analyses for dynamic kernels.

Weaknesses:

* Although the main goal of this paper is to improve the conditions made in [Chizat et al. (2019)], it still needs some more explorations, at least how this condition affects the conditions for the neural network width and the convergence rate.

* Following the previous comment, the authors should also derive the convergence rate under the improved conditions.

* The authors only consider the general NTK model, it would be better to consider the applications to certain neural network models such as multi-layer ReLU networks.

* The comparison with [Cao and Gu, 2019] should be added, in which the authors also prove a bound on the linearization approximation error (see their Lemma 5.2).

Cao and Gu 2019, Generalization Error Bounds of Gradient Descent for Learning Over-parameterized Deep ReLU Networks, https://arxiv.org/pdf/1902.01384.pdf.

---

### Author Response · Authors · 2023-08-18
**Revision**

Thank you to all the reviewers for the time taken to review the paper and for your helpful suggestions. In the revision, we:

* Added a flowchart for the proof in Figure 1, and a proof sketch in Section A.2 (suggested by Reviewer 7ztJ)

* Defined $\kappa$ outside of Proposition 1.1. (suggested by Reviewer 7ztJ)

* Added Section 2 which validates the assumptions by showing how they apply to neural networks. Note the assumptions are inherited from Chizat et al., which also has a discussion of the assumptions (suggested by Reviewers t2p5 and 7ztJ)

* Clarified in the proof sketch what the new elements are (suggested by Reviewer 7ztJ). The main challenge is not establishing the continuity and operator-boundedness guarantees. Instead, the main challenge is that we provide a new, general theorem bounding the difference between product integrals.

* Added comparison to [Cao and Gu, 2019] (suggested by Reviewer t2p5)

* Changed the title to "Tight conditions for when the NTK approximation is valid"

Please see the revision. Modifications to the text are in blue.

---

### Decision · Action_Editors · 2023-10-07

**Recommendation:** Accept with minor revision

**Comment:**

The paper is reviewed by three experts. Although two of the reviewers are positive about the paper, one reviewer raised concerns which authors only partially addressed.

In particular,  the reviewer would like to see a discussion on global convergence, which I think authors can briefly mention in the final version (does not have to be a technical statement if it is beyond the scope).

I think the paper will be a good addition to TMLR and its contributions  will be appreciated by its audience.

**Audience:**

The paper studies a very popular neural network analysis tool, NTK approximation, and tightens it. The paper is very interesting for the theoretical audience of TMLR.

**Claims And Evidence:**

This paper provides a tightened NTK approximation analysis of Chizat & Bach 2019 based on time rescaling. The results derived by the authors are tight as they match with the corresponding lower bound.

The paper is well-written and claims and statements are supported by rigorous proofs.

---

> ### Author Response · Authors · 2023-11-06
> **Convergence**
>
> Thank you. In the camera-ready we have added a remark about convergence directly after Lemma 2.1.